# Transformable topological mechanical metamaterials

D. Zeb Rocklin[1,2,3], Shangnan Zhou[1], Kai Sun[1] & Xiaoming Mao[1]

Mechanical metamaterials are engineered materials whose structures give them novel mechanical properties, including negative Poisson's ratios, negative compressibilities and phononic bandgaps. Of particular interest are systems near the point of mechanical instability, which recently have been shown to distribute force and motion in robust ways determined by a nontrivial topological state. Here we discuss the classification of and propose a design principle for mechanical metamaterials that can be easily and reversibly transformed between states with dramatically different mechanical and acoustic properties via a soft strain. Remarkably, despite the low energetic cost of this transition, quantities such as the edge stiffness and speed of sound can change by orders of magnitude. We show that the existence and form of a soft deformation directly determines floppy edge modes and phonon dispersion. Finally, we generalize the soft strain to generate domain structures that allow further tuning of the material.

[1] Department of Physics, University of Michigan, Ann Arbor, Michigan 48109-1040, USA. [2] Laboratory of Atomic and Solid State Physics, Cornell University, Ithaca, New York 14853-2501, USA. [3] School of Physics, Georgia Institute of Technology, Atlanta, Georgia 30332, USA. Correspondence and requests for materials should be addressed to D.Z.R. (email: dzr3@cornell.edu) or to X.M. (email: maox@umich.edu).

The emergence of mechanical stability is a central theme in many branches of condensed matter physics, ranging from jamming of granular matter[1,2] to strain stiffening of biopolymer networks[3–5] and structural phase transitions in crystals[6]. A universal language, which dates back to J. C. Maxwell, to characterize mechanical stability in various systems is based on frame models, which contain rigid struts connected by free-hinges[7,8]. The simple rule for the emergence of mechanical stability in these frames is $\langle z \rangle = 2d$, where $\langle z \rangle$ is the average coordination number (number of struts that meet at a hinge) and $d$ is the spatial dimension. Because each hinge as a point-like object in the model contributes $d$ degrees of freedom and each strut contributes one constraint, structures satisfying $\langle z \rangle = 2d$, now called Maxwell lattices[9], contain just enough constraints for all the degrees of freedom and thus are at the verge of mechanical instability.

What is central to the understanding of mechanical stability is the concept of floppy modes, which are normal modes of deformation that do not stretch or compress any strut and only involve rotations at the hinges in the frame. In general these modes emerge when $\langle z \rangle < 2d$ and lead to instabilities, for example, flow of granular matter. A particularly interesting case is the generation of floppy modes by cutting a finite piece from an infinite Maxwell lattice. Because the hinges on the boundary have $z < 2d$, the finite system must have as many floppy modes as the length of the boundary. Recent studies reveal that these floppy modes exhibit very rich physics. They can either appear as plane wave modes that penetrate through the bulk[2], or appear as modes localized on the edge[10]. In particular, Kane and Lubensky found that these floppy modes can reside on one side of a finite lattice, leaving the other side with no floppy modes and that this is controlled by the topology of the phonon bands[11]. This is analogous to quantum topological states of electrons, such as the quantum Hall effect and topological insulators, where edge modes are determined by the topology of the electron bands[12,13]. The phenomenon of floppy modes concentrating on one side of a lattice, called topological polarization, is a property of the lattice that is protected by topology, so it is highly robust against disorder and noise. Recent works have considered topological states of matter in a wide variety of mechanical and acoustic networks, including not only just mechanical frames[14–18] but also biological microtubules[19], coupled pendula[20], gyroscopes[21,22],

acoustic resonators[23], origami/kirigami[24] and cogs with coupled orientations[25]. Because low-energy modes often dominate the mechanical response of a structure, the rich spectrum of floppy modes in Maxwell lattices provides a great opportunity to design mechanical metamaterials in which novel mechanical responses can be programmed.

Here we show that, interestingly, simple operations that cost little energy can be utilized to induce transitions that change the topological polarization in a structure, analogous to the change of topological states in the quantum Hall effect through the change of the magnetic field. This leads to a design principle for mechanical metamaterials that can be easily and reversibly transformed between states with dramatically different properties, and we use example lattices to illustrate this (see Supplementary Video). Recently there have been many interesting proposals for tunable mechanical metamaterials[26–35]. What is unique to this design is that, first, the unusual asymmetric properties in the different states are protected by topological invariants of the phonon bands and thus the system is more robust against possible wear from repeated transformations, and second, the operation is based on a soft deformation of the structure that uniformly twists the angles at the hinges throughout the system and thus costs little energy and involves little stress. Hence, we refer collectively to such systems as transformable topological mechanical metamaterials (TTMM). We further show that any structure exhibiting such a uniform soft deformation can be classified via whether this deformation is predominantly shear (shear dominant) or dilation (dilation dominant), two regimes with sharply different properties. This general classification provides a guideline for the creation of new TTMMs. We discuss possible experimental systems that can be used to fabricate TTMMs. We also discuss how these soft deformations can be used to create novel domain structures, in which topological polarizations can be tuned locally, leading to versatile control of stiffness along edges and domain walls.

## Results

**Topological transitions induced by uniform soft twistings.** We start our discussion of the topological transitions using the example of the deformed kagome lattice, which is the first two-dimensional lattice that was shown to exhibit topological

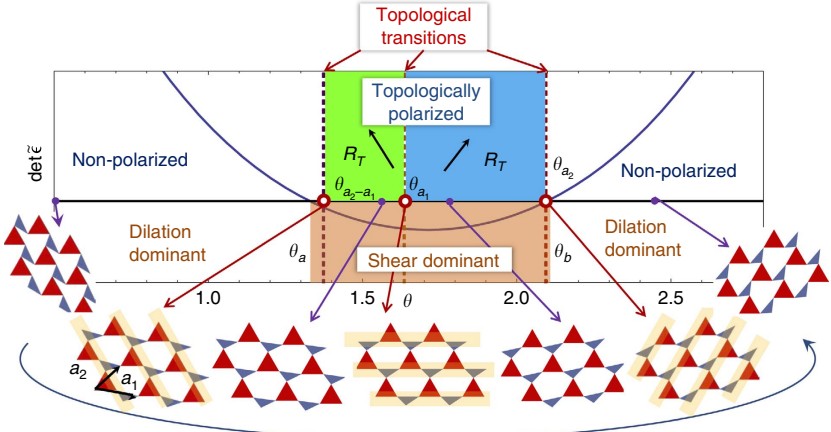

Transforming lattice with uniform soft twisting $\theta$

**Figure 1 | Topological transitions of a deformed kagome lattice by uniform soft twisting.** Two types of triangles (red and blue) are connected by free hinges at their corners, forming a deformed kagome lattice with primitive vectors $\mathbf{a}_1$, $\mathbf{a}_2$. The angle $\theta$ between the triangles defines the twisting coordinate. The blue curve shows $\det \tilde{\epsilon}$ (defined in equation (1)) as a function of $\theta$. The 3 white dots on the $\theta$ axis represent three critical angles ($\theta_{\mathbf{a}_2 - \mathbf{a}_1}$, $\theta_{\mathbf{a}_1}$ and $\theta_{\mathbf{a}_2}$) where sides of the triangles form straight lines (yellow stripes on the lattices) and topological polarization $\mathbf{R}_T$ (shown as black arrows above the axes) changes.

polarization[11]. This lattice is constructed by connecting rigid triangles with free hinges as shown in Fig. 1 (the term deformed refers to the fact that this lattice consists of triangles of shapes that differ from those in the regular kagome lattice, and does not mean the lattice is stressed). This structure is a Maxwell lattice because each triangle can be viewed as three struts and thus each hinge connects four struts, so $z = 2d$. The lattice in Fig. 1 has chosen side lengths $(1, 1, 1)$, $(0.72, 1, 0.57)$ for the two triangles.

The topological polarization $\mathbf{R}_T$, as introduced in ref. 11, is a vector that represents topological invariants of the phonon modes in the first Brillouin zone of the lattice, and is determined by the lattice geometry. When $\mathbf{R}_T = 0$ the lattice has floppy modes on all edges, but when $\mathbf{R}_T \neq 0$, the direction it points towards is the edge that gains extra floppy modes, whereas the opposite edge lacks floppy modes.

As first shown by Guest and Hutchinson[36], all two-dimensional periodic Maxwell lattices must exhibit at least one homogeneous deformation (that is, all repeating units deform in the same way) that is soft (no stretching or compression of struts and only rotations at hinges), regardless of the specific unit cell geometry (more detailed discussion in Supplementary Note 2). The deformed kagome, as a Maxwell lattice, must exhibit such a mode, as shown in Fig. 1. We find that as the lattice follows this nonlinear soft deformation the topological polarization $\mathbf{R}_T$ can change, which dictates that the edge floppy modes have to move from one edge to the opposite edge at the transition. In the following discussion, we call such uniform floppy deformations uniform soft twistings, and we use a bond angle $\theta$ (Fig. 1) as the coordinate to label states along the path of the uniform soft twisting.

As shown in Fig. 1 the deformed kagome lattice experiences three topological transitions at $\theta_{\mathbf{a}_2 - \mathbf{a}_1}$, $\theta_{\mathbf{a}_1}$, $\theta_{\mathbf{a}_2}$ (labelled by the associated lattice directions in terms of the primitive vectors of the lattice marked in Fig. 1), with their order dependent on lattice geometry. At twisting angles below the first or above the last transition, the lattice has $\mathbf{R}_T = 0$ where floppy edge modes reside on all edges. As $\theta \to \theta_{\mathbf{a}_2 - \mathbf{a}_1}^-$ edge modes on the bottom edge penetrate deeper and deeper into the bulk and eventually become bulk modes (with zero decay rate) at $\theta = \theta_{\mathbf{a}_2 - \mathbf{a}_1}$. Upon further increasing $\theta$, these modes transform into edge modes on the top edge, doubling the number of floppy modes there. This evolution of floppy modes as $\theta$ increases is illustrated in Fig. 2a–c. The transitions at $\theta_{\mathbf{a}_1}$ and $\theta_{\mathbf{a}_2}$ are of the same nature, where floppy modes shift from certain edges to edges on the opposite side of the system. As $\theta$ crosses the three critical angles $\theta_{\mathbf{a}_2 - \mathbf{a}_1}$, $\theta_{\mathbf{a}_1}$, $\theta_{\mathbf{a}_2}$, the change of $\mathbf{R}_T$ follows $0 \to (\mathbf{a}_2 - \mathbf{a}_1) \to \mathbf{a}_2 \to 0$, so the two regimes $\theta_{\mathbf{a}_2 - \mathbf{a}_1} < \theta < \theta_{\mathbf{a}_1}$ and $\theta_{\mathbf{a}_1} < \theta < \theta_{\mathbf{a}_2}$ have distinct nonzero $\mathbf{R}_T$ and are topologically polarized. The transitions at $\theta_{\mathbf{a}_2 - \mathbf{a}_1}$, $\theta_{\mathbf{a}_1}$, $\theta_{\mathbf{a}_2}$ are called topological transitions because a topological index $\mathbf{R}_T$ changes its value across them. At a transition, edges of the triangles form straight lines along a particular direction, which is intimately related to the rise of bulk floppy modes at the transition. As discussed in refs 9–11, straight lines in the bulk allow states of self stress (ways to distribute internal stress without net forces on any parts) such that floppy bulk modes can arise. This shares interesting similarities with the quantum Hall effect, where the system must pass through a metallic state as it transforms between insulating states of different topological indices. It is also reminiscent of the process whereby the topological polarization of a one-dimensional Maxwell frame lacking translational invariance may be altered gradually by a soliton, a nonlinear local deformation[14].

These transitions lead to a dramatic change in the edge stiffness because an edge becomes rigid as it loses floppy modes. We perform conjugate-gradient minimization calculations of the response to a point force on one edge of a lattice with other edges held fixed (for details see Supplementary Note 3), and find that the edge stiffness increases by orders of magnitude as floppy modes leave the edge (Fig. 2d).

**Classification of structures with soft twistings.** The example of the deformed kagome lattice provides a simple design for mechanical metamaterials that are transformable between states of sharply different but topologically protected mechanical properties. To explore new structures that exhibit such topological transitions, here we also study the general classification of structures that exhibit uniform soft twistings.

To achieve this general classification, we first analyse the consequences of the uniform soft twisting, which (around a given state) can be written in terms of the left Cauchy–Green strain tensor

$$\tilde{\epsilon} = \begin{pmatrix} \tilde{\epsilon}_{xx} & \tilde{\epsilon}_{xy} \\ \tilde{\epsilon}_{xy} & \tilde{\epsilon}_{yy} \end{pmatrix}, \tag{1}$$

which is homogeneous in space and is a function of $\theta$. As proved in Supplementary Note 4, utilizing the fact that any elastic deformation in flat space must have zero curvature, the existence of the zero energy uniform deformation $\tilde{\epsilon}$ leads to two families of spatially varying floppy modes described by strain tensors

$$\begin{aligned} \epsilon_+ (\mathbf{r}) &= \tilde{\epsilon} f_+ (x + \lambda_+ y), \\ \epsilon_- (\mathbf{r}) &= \tilde{\epsilon} f_- (x + \lambda_- y), \end{aligned} \tag{2}$$

where $\mathbf{r} = (x, y)$ is the coordinate, $f_\pm (w)$ are two arbitrary scalar functions and $\lambda_\pm$ are two constants determined by $\tilde{\epsilon}$

$$\lambda_\pm = \left( \tilde{\epsilon}_{xy} \pm \sqrt{-\det \tilde{\epsilon}} \right) / \tilde{\epsilon}_{xx}, \tag{3}$$

where $\det \tilde{\epsilon} = \tilde{\epsilon}_{xx} \tilde{\epsilon}_{yy} - (\tilde{\epsilon}_{xy})^2$ is the determinant of $\tilde{\epsilon}$.

The characteristics of these floppy modes are dictated by the sign of $\det \tilde{\epsilon}$, which distinguishes two different regimes: the dilation dominant regime, $\det \tilde{\epsilon} > 0$ and the shear dominant regime, $\det \tilde{\epsilon} < 0$. Because $\det \tilde{\epsilon}$ is independent of the choice of coordinates, it measures an intrinsic property of the uniform soft twisting. In addition, structures in the dilation dominant regime are necessarily auxetic[31] because they have $\tilde{\epsilon}_{xx} \tilde{\epsilon}_{yy} > 0$, which gives a negative Poisson's ratio.

In the dilation dominant regime ($\det \tilde{\epsilon} > 0$), the floppy modes are edge modes localized on all edges of the system. This conclusion is transparent after we decompose the two arbitrary functions $f_\pm$ into Fourier series $f_\pm (w) = \sum_k \phi_\pm (k) e^{ikw}$, so that the functions in equation (2) turn into

$$f_\pm (x + \lambda_\pm y) = \sum_k \phi_\pm (k) e^{ikx + i\lambda_\pm ky}. \tag{4}$$

For any real number $k$, along the $x$ direction, the exponential factor $e^{ikx}$ describes a plane wave with wave number $k_x = k$. However, along $y$, because $\lambda_\pm$ is complex for $\det \tilde{\epsilon} > 0$, its imaginary part, $\text{Im}\lambda_\pm$, yields a factor $e^{-\kappa y}$ with $\kappa = k \, \text{Im}\lambda_\pm$, so that the amplitude of this deformation decays exponentially along the $y$ axis. If the system has an open edge parallel to the $x$ axis, this is a plane wave along the edge whose amplitude decays exponentially from the edge into the bulk of the system, that is, an edge mode with zero sound velocity. The decay rate for this edge mode is proportional to the wavevector, $\kappa \propto k$. Because the $x$ direction here is chosen arbitrarily, the same conclusion applies to arbitrary edge directions and thus floppy modes arise on all edges. Because the elastic theory shows no bulk floppy modes, the bulk is in general rigid and has no floppy mode except the uniform soft twisting, which we assumed from the beginning. One special case in the dilation dominant regime, the twisted kagome lattice, was discussed in ref. 10, where the uniform soft twisting is a pure dilation $\tilde{\epsilon}_{xy} = 0$ and $\tilde{\epsilon}_{xx} = \tilde{\epsilon}_{yy}$. For that special case, the system has

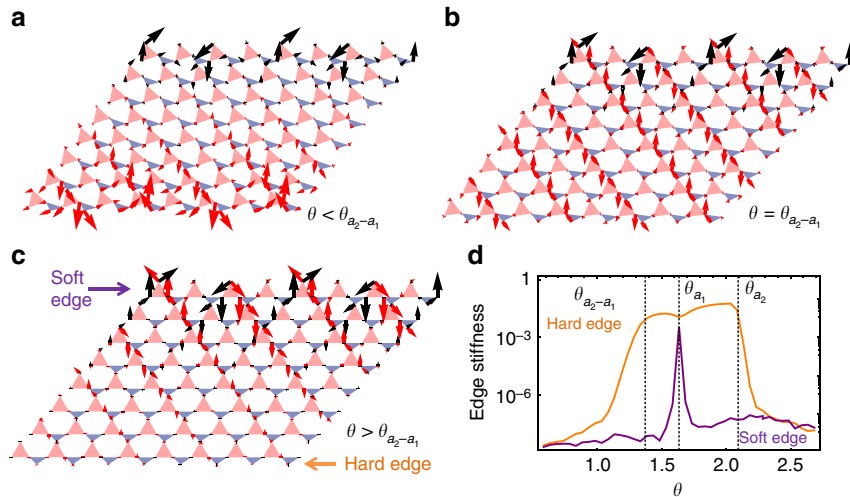

**Figure 2 | Transformation of edge modes and stiffness.** (**a**–**c**) Evolution of a pair of floppy modes (red and black arrows) as the example deformed kagome lattice shown in Fig. 1 traverses its soft twisting coordinate $\theta$ across the critical angle $\theta_{a_2 - a_1}$, where the lattice develops a topological polarization. Periodic boundary conditions are applied to left-right edges and open boundary conditions to top-bottom edges. (**d**) Numerical results for the dramatic change of stiffness of the hard (bottom) edge against local displacements as $\theta$ changes and floppy modes move to the opposite edge. Data is from a $60 \times 60$ generic kagome lattice with the structure shown in (**a**) with free hinges and fixed boundaries except the measurement edge (see Supplementary Note 1).

**Table 1 | General classification of lattices with uniform soft twistings.**

| Soft twisting characteristic | Shear dominant $\det \tilde{\epsilon} < 0$ | | Dilation dominant $\det \tilde{\epsilon} > 0$ | |
|---|---|---|---|---|
| Spatially varying floppy modes | $k_y = \lambda_\pm k + O(k^2)$ with $\lambda_\pm \in \mathbb{R}$ | | $k_y = \mathrm{Re}(\lambda_\pm) k + i\,\mathrm{Im}(\lambda_\pm) k + O(k^2)$ | |
| Bulk phonon spectra | Vanishing speed of sound in two directions | | Positive speed of sound in all directions | |
| | | $z = 2d$ | $z > 2d$ | $z = 2d$ | $z > 2d$ |
| Floppy edge phonons (FEP) — Example lattices | | | | | |
| Floppy edge phonons (FEP) — Frequency | | $\omega = 0$ | | $\omega = 0$ | $\omega = O(k^2)$ |
| Floppy edge phonons (FEP) — Decay rate | | $O(k^2)$ Can be 0 in some cases (bulk phonon) | FEP NOT guaranteed to exist | $O(k)$ | $O(k)$ |
| Floppy edge phonons (FEP) — FEP features | | Can be topologically polarized | | FEP appear in pairs at opposite edges and can be described by generalized conformal transformations | |

The spatially varying floppy modes are expressed in terms of the wave number in the *y* direction when a plane wave of wave number *k* propagates in the *x* direction (see discussions after equation (4)). The bulk phonon spectra show example phonon frequency contour plots (darker colour for lower frequency) as a function of $k_x$, $k_y$. The example lattices are shown as rigid polygons (triangles or parallelograms) connected by free hinges at their corners[45], and they can be directly mapped into strut-hinge frames by replacing the triangles by three connected struts on their edges and the parallelograms by five connected struts with four on edges and one on the diagonal to make them rigid. Thus the structures consisting of triangles (deformed kagome lattices as defined the text) have $\langle z \rangle = 4 = 2d$ and the structures consisting of parallelograms (deformed checkerboard lattice) have $\langle z \rangle = 5 > 2d$.

an emergent conformal symmetry and the floppy edge modes are conformal deformations. As we prove here, the same qualitative properties shall always arise as long as $\det \tilde{\epsilon} > 0$.

For the shear dominant regime $\det \tilde{\epsilon} < 0$, the floppy modes are bulk plane waves along two special directions $k_y = \lambda_+ k_x$ and $k_y = \lambda_- k_x$, as can be seen directly from equation (4) with real valued $\lambda_+$ and $\lambda_-$. For bulk sound waves along these two special directions, the sound velocity vanishes, which is the key acoustic

signature of the shear dominant regime. On the edge of the system, our general elastic theory neither requires nor prevents the existence of floppy edge modes, implying that the fate of the edge is not universal and relies on the architecture of the lattice. Generally in a solid, surface or edge sound waves, known as Rayleigh waves, could arise and the frequencies of these Rayleigh waves are lower than those of waves in the bulk (surface waves can also have frequencies located in a phonon bandgap, but

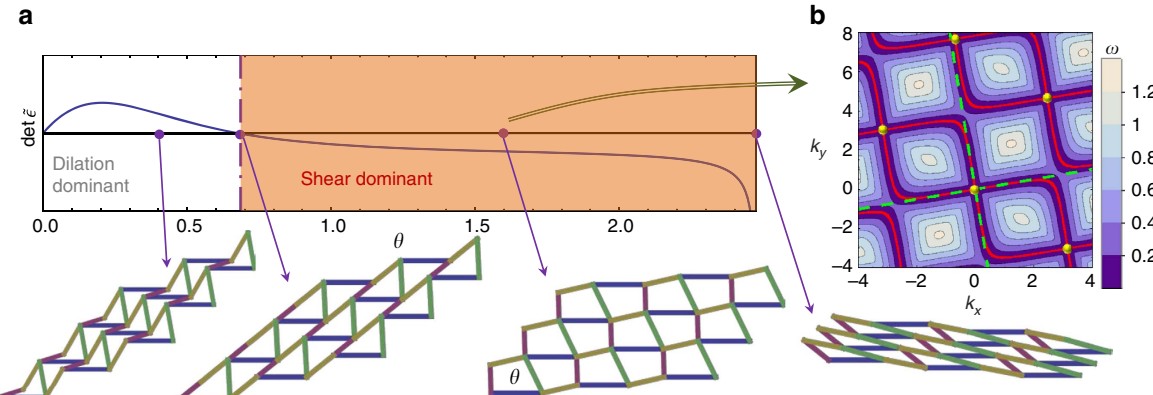

**Figure 3 | Evolution of a non-topological lattice and its bulk floppy modes upon uniform soft twisting.** (**a**) Uniform soft twisting of a deformed square lattice constructed of four struts of different lengths (four different colours), with each primitive unit cell contains two hinges. The spatially varying floppy modes in the deformed square lattice when $\det \tilde{\epsilon} < 0$ are bulk instead of floppy edge phonons. Unlike the deformed kagome lattice of Fig. 1, the shear dominant (orange) region does not contain topologically polarized lattices. (**b**) An example of the phonon frequency contour plot (as a function of momenta $k_x$, $k_y$), where the zero frequency phonon modes are shown in red, the two green dashed lines show the two zero speed of sound directions $(1, \lambda_{\pm})$ given by equation (3), and the yellow dots show reciprocal lattice sites.

because we only focus on low-frequency phonon modes, this case will not be considered here[37]. Because their frequencies are lower than the bulk ones, including the floppy bulk plane waves with zero sound velocity, these surface waves will also be soft and have zero sound velocity. At long wavelengths (small $k$), these floppy edge modes have decay rate $\kappa \sim k^2$ and penetrate much deeper into the bulk, in comparison with the floppy edge modes in the dilation dominant regime discussed above, which have $\kappa \sim k$.

We summarize this classification in the table in Table 1. It is worth pointing out that the analysis here is not limited to frames. The only assumption we make is the existence of a soft uniform deformation that only costs elastic energy to $O(\epsilon^3)$, as opposed to normal elastic deformations which cost energy $O(\epsilon^2)$. As we show in Supplementary Note 4, the elastic energy of these resulting spatially varying floppy modes also have $E \sim O(\epsilon^3)$. In this sense they are called floppy modes or soft modes. Knowledge of the microscopic structure provides more information on these floppy modes. In particular, floppy modes of exactly zero energy (zero frequency) are guaranteed to exist in finite Maxwell lattices, as we discussed earlier. In contrast, if a structure with $\langle z \rangle > 2d$ has a uniform soft twisting (by fine-tuning of its architecture, such as the example of the deformed checkerboard lattice in Table 1), these modes are floppy $E \sim O(\epsilon^3)$ but not guaranteed to be zero energy. Although we focus here on elastic and low-frequency properties of the material, past work has that twisting of a regular kagome lattice induces bandgaps and other interesting features in the full phonon dispersion[35].

This classification sheds light on topological states of a structure. One straightforward relation is that there can be no stiff edge in the dilation dominant regime. This is because all edges must have floppy modes in this regime, as we have shown above. For example, in Fig. 1 the deformed kagome lattice enters the shear dominant regime shortly before the first topological transition and returns to it after the last topological transition returns it to an unpolarized state. The phenomenon of stiff edges only shows up in the polarized configurations, $\theta_{a_2 - a_1} < \theta < \theta_{a_2}$.

Furthermore, there are lattices that exhibit a transition between dilation dominant and shear dominant regimes but never exhibit a topological phase. One example is the deformed square lattice with two hinges and four struts of different length per unit cell (as shown in Fig. 3 where strut lengths (1.3, 1, 0.9, 0.7) was used). This structure also has one uniform soft twisting, which changes the angle $\theta$ uniformly. As $\theta$ increases, the system undergoes one

transition from the dilation dominant regime to the shear dominant one. Agreeing with our elastic theory, the dilation dominant regime shows a rigid bulk and soft edges, while the shear dominant regime has floppy bulk modes. Interestingly, in contrast to the deformed kagome lattice, the deformed square lattice shows no floppy edge modes. Instead it has bulk modes with exactly zero energy. These floppy bulk modes follow the predicted directions $(1, \lambda_{\pm})$ at small $k$, but deviate at larger $k$ (zero frequency lines in Fig. 3 are curved). Thus this lattice provides a different class of structures that exhibit uniform soft twisting.

**Creation of domain structure**. We now show that the critical configurations of a Maxwell lattice may be used to generate a new family of domain structures that provide local control of the topological polarization and hence mechanical response. As we've discussed, twisting a Maxwell lattice through a topological transition moves zero modes into the bulk as they pass from one edge to the other, altering the edge stiffness. Interestingly, uniquely at the critical point, these floppy modes are in the bulk, and there is an opportunity to use them to systematically alter the structure of any Maxwell lattice, as discussed in detail in Supplementary Note 5. Such modes were previously used in the special case of the standard kagome lattice to take edge zero modes into the bulk[10] and to alter the finite-frequency wave structure[35]. As shown in Fig. 4, when the twisting angle reaches $\theta_{a_1}$, the point at which $a_1$ reaches maximum, ($\theta_{a_2}$ corresponds to a maximum of $a_2$ and $\theta_{a_2 - a_1}$ to the third lattice direction $a_1 - a_2$), either slightly increasing or slightly decreasing the twisting angle $\theta$ results in the decrease of $a_1$. The same primitive vector $a_1$ can be reached by two choices of twisting angle $\theta_t$ and $\theta_b$ which are above and below $\theta_{a_1}$. Thus, two domains of the deformed kagome lattice at twisting angle $\theta_t$ and $\theta_b$ may be joined together at a domain wall interface as shown in Fig. 4c. Hence, from the homogeneous critical configuration one may choose for each of the $N_y$ rows separately one twisting direction or the other, resulting in a domain wall structure that then passes on a fixed course (represented as a blue line in Fig. 4) as the lattice twisting is continued until another homogeneous critical point is reached (full traversal of this path requires tiles to pass through or over one another). Note that the structure cannot jump from one path to another path (different blue lines). The only way to change the

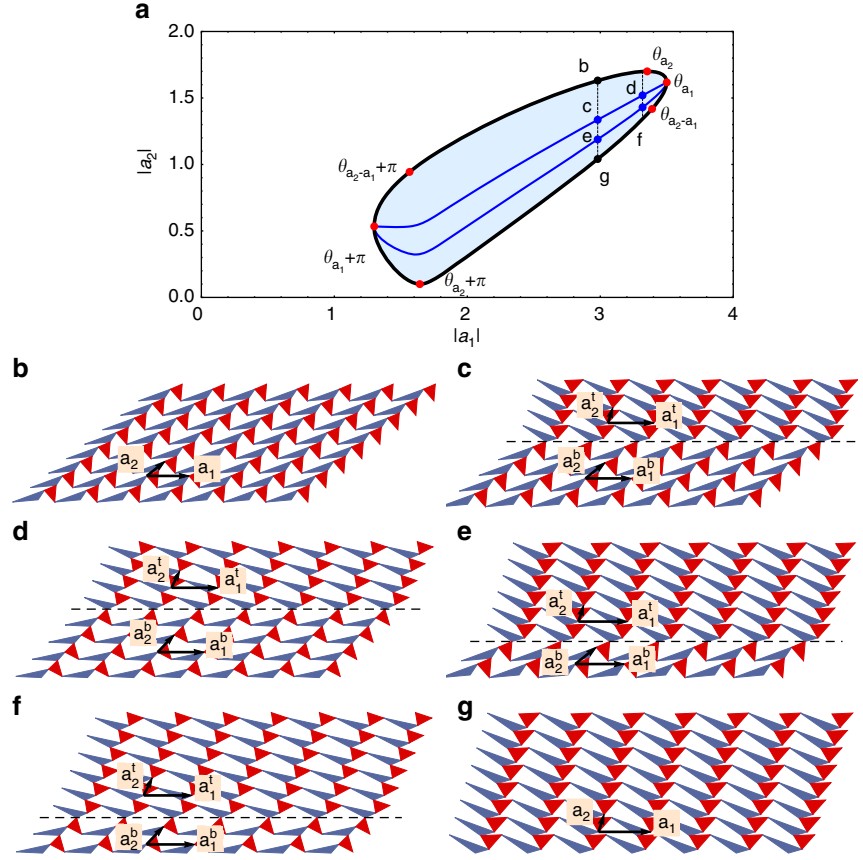

**Figure 4 | Transformation-induced domain structure.** The nonlinear uniform soft twisting of a deformed kagome lattice alters the lengths of the two lattice vectors $\mathbf{a}_1$, $\mathbf{a}_2$ as indicated by the thick black curve of **a**. However, as discussed in the main text, at critical angles such as $\theta_{\mathbf{a}_1}$ it is possible to create distinct domains within which two sections of the material undergo the uniform soft twisting depicted in Fig. 1 in opposite directions. This leads to a family of new twisting modes such as the blue lines shown. Choosing the proper domain structure allows any set of material strains in the shaded region. In configuration (**c**), the top and bottom domains correspond, respectively, to the uniform configurations (**b**,**g**), with continuity at the interface allowed by the common length of the primitive vector $\mathbf{a}_2$. Configuration (**c**) may be transformed into configuration (**d**) by the soft twisting, but to achieve either a uniform configuration or configurations (**e**,**f**), which have different distributions of domains, the material must pass through a critical angle.

number of rows in each domain is to return to the critical lattice at $\theta_{\mathbf{a}_1}$.

By choosing the fraction of the system subject to one twisting or the other, one can tune the effect on the average lattice primitive vectors, resulting in a new nonlinear soft strain, which we call domain soft twisting. The domain soft twistings can be used to generate a desired mechanical response. For example, choosing a path such that $|\mathbf{a}_2|$ remains nearly unchanged while $|\mathbf{a}_1|$ changes by a significant fraction allows the material to undergo stress-free deformations that dilate significantly along one lattice direction while leaving the other unchanged (and also undergoing some general shear). We emphasize that this response is not set by the material but tuned by the domain structure it is placed in at the critical point.

Changing the domain structure through the domain soft twistings has very interesting effects on the edge floppy modes of the Maxwell lattices. To demonstrate this, we show the spatial extent of zero modes, obtained by directly calculating the zero modes of a finite Maxwell lattice and measuring the amplitude of zero modes projected into every lattice site. The total amplitude $\phi_i$ of zero modes $\{\mathbf{u}_0^j\}$, which projects into a site $i$'s two degrees of freedom is

$$\phi_i = \sum_j \left(\hat{\mathbf{x}}_i \cdot \mathbf{u}_0^j\right)^2 + \left(\hat{\mathbf{y}}_i \cdot \mathbf{u}_0^j\right)^2, \qquad (5)$$

where $j$ labels zero modes, and $\phi_i$ varies from 2 for completely free particles in two dimensions to 0 for completely pinned particles. The spatial extent of zero modes $\phi_i$ is an indicator of the local 'floppiness', because stress-free spatially localized deformations such as the edge indentation considered in Fig. 1d require spatially localized zero modes, which is measured by $\phi_i$.

We show the effect of the domain soft twisting on the edge floppy modes in Fig. 5. We examine a collection of domain structures created from the same lattice, with the domain twisting angles shown in the top row, and the effect on the edge modes is shown in Fig. 5b–g where green indicates area with high $\phi_i$ and thus higher floppiness.

For a homogeneous unpolarized state, the zero modes are distributed roughly evenly around the edge, as shown in Fig. 5a. Upon twisting the lattice to the critical angle $\theta_{\mathbf{a}_1}$ and creating top and bottom domains only one of these domains is polarized along the $\mathbf{a}_1$ direction, as shown in Fig. 5b. This can be used to create edge stiffnesses that change along the length of the edge in desired ways. For example, in Fig. 5c the middle domain of the left edge has been given zero modes, creating a soft spot in an otherwise stiff surface.

The domain structure can also be used to enlarge the crystal basis of a cell and reduce its symmetry, similar to the 'multiple folding mechanisms' considered previously in regular kagome lattices[35]. Rather than creating large domains, one may twist each row oppositely from its two neighbours, as shown in Fig. 5d. The

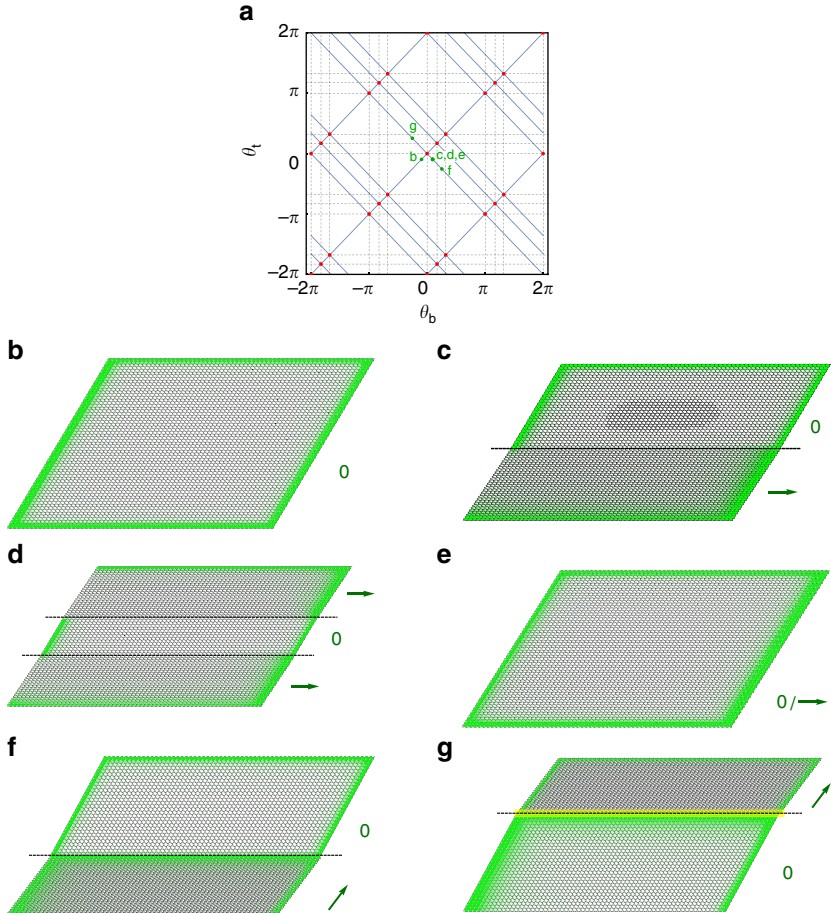

**Figure 5 | Manipulating edge modes via domain structure.** (**a**) Blue lines depict allowed configurations of a deformed kagome lattice, parametrized by the twisting angles ($\theta_t$, $\theta_b$) of two types of domains. At critical (red) points the two domains may be twisted in opposite directions, and continuing this twisting allows them to be separately polarized, passing through dashed lines. Green points denote the configurations shown in the remaining panels, wherein the spatial extent of the lattice's zero modes is shown in green. (**b**) Homogeneous, unpolarized lattice, with zero modes evenly distributed along the boundaries. (**c**) Lattice in which the domain below the dashed line has polarization pointing to the right, so that there are no zero modes on the left edge in the lower domain. In states such as (**d**), the domain structure may be chosen, so that only desired portions of the left edge (in this case, the middle of the edge) have zero modes and are deformable, while the rest are rigid. In (**e**), alternating rows are twisted in opposite directions, as though each row is its own micro-domain, such that half of the zero modes originating on the left wall have been shifted to the right. In (**f**), the configuration from (**c**) is twisted further, so that the bottom edge undergoes a second topological transition (passing through the dashed line in the top diagram), shifting zero modes onto the domain wall itself. In (**g**), the order of the domains is reversed as compared with (**f**), so that the domain wall acquires a line that can support stress (a self stress) rather than a zero mode.

particularly interesting consequence for having these repeating domains is that it provides us with a method to shift a fraction of the edge zero modes from edge to edge. In the example shown in Fig. 4d, half of the edge floppy modes that were originally on the left edge are shifted to the right edge, resulting in a lattice that is 'half' polarized by its original unit cell (though it still has integer polarization as defined consistently with its now-larger unit cell). In this way, by choosing a finite fraction of domains to twist in one direction one may robustly decrease the edge stiffness by a controlled fraction, rather than the exponential loss of edge stiffness shown previously for fully uniform polarization. Thus, the domain structure selected at the critical angle may be used to 'continuously' tune the edge stiffness. Twisting away from the critical angle fixes the domain structure and hence provides topological protection for this stiffness.

Furthermore, different domains can undergo additional topological polarizations separately, as shown in Fig. 5, which depicts a path of allowed pairs of domain-specific twisting angles $\theta_t$, $\theta_b$ as one or the other passes through topological transitions (dashed lines). In this way, one may create topological

polarizations pointing towards domain walls, resulting in domain walls that contain zero modes and hence may be used as channels to convey motion across an otherwise rigid bulk, as shown in Fig. 5e. Hence, we have shown a way to create in an initially homogeneous lattice zero modes as desired of the sort first identified by Kane and Lubensky[11] in static interfaces between dissimilar lattices. That work also identified structures known as *self stresses* along such domain boundaries, which we may similarly create as shown in Fig. 5(VI) by choosing a different spatial order for the domains. It has been shown that such self stresses are capable of bearing external stress independent of the rest of the structure[16]. Hence, the chosen domain structure allows the creation of channels across the bulk of the material that convey both motion and stress. Other works have engineered zero modes and self stresses via static defects[15] rather than exploiting nonlinear domain soft twistings of the materials.

This method of creating domains should be fully extendable to analogous three-dimensional (3D) structures. A domain plane in such a structure would require that that the amplitudes of and the angle between the two primitive vectors in the plane be equal

across the domain. However, 3D Maxwell lattices have three Guest modes, and hence these three requirements should result in a unique path through configuration space for a given set of domains. It has recently been shown that 3D Maxwell lattices can be topologically polarized and have other nontrivial properties[18].

## Discussion

In summary, we have examined TTMMs, a broad class of systems characterized by their uniform soft elastic deformations. As we have shown, this soft deformation can be activated to reversibly tune, without the application of significant stress, between states with dramatically different edge stiffnesses, bulk speeds of sound and location and spatial extent of floppy edge modes. All of these properties are robust, with some protected by the system's topological state and others guaranteed by the form of the soft elastic deformation. Domain structures may be implemented at certain lattice configurations, allowing generalized nonlinear modes and localized control of elastic properties through the change of topological states in domains. These materials reveal a deep connection that endures between structure and function even as both are altered, and have broad potential applications. Realization of the full phenomenology, including inserting domain wall structures, requires flexible hinging motions capable of undergoing large deformations without significant resistance and fine manipulation of the edge or bulk.

In the Supplementary Video, we demonstrate a dramatic change of the edge stiffness at the topological transition using a macroscopic prototype built with K'nex, hard plastic parts joined by literal hinges. More generally, because topological transitions discussed here only require periodic arrangements of rigid building blocks connected by relatively soft junctions (bending stiffness at junctions can be treated as a perturbation, which gradually increase elastic energy of the floppy surface modes), TTMMs may be realized in a broad range of experimental systems. For example, the lattices studied in this Article can be produced via 3D printing or lithography[16,38], in which hinges can be approximated by thin joints between struts or polygons. The TTMMs may also be self-assembled[39,40] from polygons at the microscale. If tip-to-tip binding can be realized between polygons through some form of directional interactions such as capillary force[41,42], these lattices may also be self-assembled, and the flexibility at the tip binding sites between the polygons would exhibit very small bending stiffness.

Examples of future applications of TTMMs include materials with tunable ability to conform to a surface, a quality that affects adhesion and grip in systems as varied as rubber tires that can be switched to perform on hard or soft surfaces[43], gecko pads and nanotechnology inspired thereby[44], as well as car components that are rigid in their nominal state as load-bearing elements but become more compliant and energy-absorbing to reduce load impulse to the passenger compartment.

## Methods

Our primary methods were analytical theories aided by computer-aided symbolic analytical and numerical computations in the program Mathematica. The prototype was generated from assembling commercially available plastic parts, known as K'Nex and other simple load-bearing elements (metal rods).

**Data availability.** Further details are available in the Supplementary Notes. The datasets generated and protocols used and analysed during the current study are available from the corresponding authors on reasonable request.

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

## Acknowledgements

We thank Tom C. Lubensky, Vincenzo Vitelli and Bryan Chen for useful discussions. D.Z.R. thanks NWO and the Delta Institute of Theoretical Physics for supporting his stay at the Lorentz Institute. This work was supported in part by the ICAM postdoctoral and Bethe/KIC fellowships (D.Z.R.), the National Science Foundation, under grants DMR-1609051 at the University of Michigan (X.M.), PHY-1402971 at the University of Michigan (K.S.), and the the James Van Loo Applied Mathematics and Physics Undergraduate Support Fund (S.Z.).

## Author contributions

D.Z.R., K.S. and X.M. conceived the research. D.Z.R., S.Z. and X.M. developed the prototype and demonstration. All authors contributed to the analytical calculations and the manuscript preparation.

## Additional information

**Competing financial interests:** The authors declare no competing financial interests.

**Publisher's note**: 

