## [Peer Review File · Nature Communications]

Reviewers' comments:

Reviewer #1 (Remarks to the Author):

Dear Editor,

Please find enclosed my review of the manuscript by D. Z. Rocklin and co-authors entitled "Transformable topological mechanical metamaterials". The work falls in the general class of studies that exploit the unusual properties of the mechanical/acoustic modes supported by structured mechanical metamaterials to tailor their properties, such as edge stiffness/speed of sound.

The main findings of the manuscript are the identification of simple operations that cost very little energy, but induce transitions between topologically protected states characterized by dramatic changes in their mechanical properties. The other merit of the manuscript is to delineate a general classification of the structures that exhibit this transitions.

The manuscript is in general well-written, the arguments are well supported by the theory/simulations and SI video is very instructive and useful in supporting the claims.

The main issue is if these results are novel enough to warrant publication in Nature Communications or the manuscript is more suitable for a different publication venue. There are a number of publications in the field of topological mechanical metamaterials that describe similar topological state-engineering of related structured mechanical systems [1], [2], [3], [4] and there are also some proposals to induce tunability in these materials [5], [6] (some of these references are surprisingly not cited in the manuscript). I think the authors should set-up a clear distinction between their approach and the approaches in [5], [6], emphasising not only the different physical nature of the transition mechanisms but also the implications for potential applications (Advantages & disadvantages) of their approach. Provided that this is done in a satisfactory manner, I am inclined to recommend the publication of the manuscript in Nature Communications.

[1] Sun, K., Souslov, A., Mao, X. & Lubensky, T. C. Surface phonons, elastic response, and conformal invariance in twisted kagome lattices. Proc. Natl Acad. Sci. USA 109, 12369-12374 (2012).

[2] Kane, C. L. & Lubensky, T. C. Topological boundary modes in isostatic lattices. Nature Phys. 10, 39-45 (2014).

[3] Chen, B. G., Upadhyaya, N. & Vitelli, V. Nonlinear conduction via solitons in a topological mechanical insulator. Proc. Natl Acad. Sci. USA 111, 13004-13009 (2014).

[4] J. Paulose, B. G.-g. Chen, and V. Vitelli, Nat. Phys. 11, 153 (2015).

[5] Shan, S., Kang, S. H., Wang, P., Qu, C., Shian, S., Chen, E. R. and Bertoldi, K. (2014), Harnessing Multiple Folding Mechanisms in Soft Periodic Structures for Tunable Control of Elastic Waves. Adv. Funct. Mater., 24: 4935-4942. doi:10.1002/adfm.201400665

[6] Paulose J, Meeussen AS, Vitelli V (2015) Selective buckling via states of self-stress in topological metamaterials. Proc Natl Acad Sci USA 112(25):7639-7644.

Reviewer #2 (Remarks to the Author):

The manuscript by Rocklin et al reports examples of deformable lattices whose soft deformation modes induce topological phase transitions in the phonon spectrum.

The general area of topological mechanics is currently a 'hot topic'. The work appears to be solid, correct and generally well presented. After some necessary revisions it will certainly be ready for publication somewhere.

What is less clear to me, is what is the substantial advance this work brings to the field? As far as I can see, all concepts involved in the work (deformation induced transitions, shifting floppy modes around by deformation, cute videos with lego-like toys etc) have been presented elsewhere, as the

authors readily acknowledge.

The main contribution is then the identification of a range in parameter space in which the lattice deformations that induce topological transitions are particularly soft, and some analysis of these parameter regions. I would be very interested to hear from the authors more about why this is interesting or useful.

On the 'useful' front, the authors make some somewhat vague references to the wonders of modern material science. In my opinion the sub-field is past this being a suitable justification and more precise identification of applications is necessary.

On the interesting front, I see potential in the work, perhaps the authors could make use of the nominal softness of the deformation modes to go to entirely new regimes? New nonlinear excitations? Thermal effects enabled by softness? Any one of these (or other) concepts that could be enabled by the identification of this region of parameter space would be interesting and give the work some conceptual impact.

Without either of the above however, the work appears to me to be a valid technical analysis of specific lattices. Interesting to the specialist certainly, but less obviously of broad interest. In writing this I would like to emphasize that I feel there is potential in there and the authors certainly have the necessary command of the field to take it one step further. I would be glad to see a new manuscript with such additions.

In considering a revised manuscript, I would also recommend the authors expand their discussion of topological mechanics in the early paragraphs. The field now includes other systems, such as coupled pendula (ETH), spinning top mechanics (Harvard, Chicago), among others. The central claim the authors write in *italic* has certainly been seen in such systems, as well as other floppy mode systems, so perhaps should be de-emphasized.

Referee 1, Point 1:

“There are a number of publications in the field of topological mechanical metamaterials that describe similar topological state-engineering of related structured mechanical systems [1], [2], [3], [4] and there are also some proposals to induce tunability in these materials [5], [6] (some of these references are surprisingly not cited in the manuscript)”

The references not previously included are those given as [3], [4], and [5]. We agree that it is important to include these references. [3,4] are both cited for examples of topological states in mechanical frames, and [5] is cited as an example of tunable mechanical metamaterials. In addition, [3] is now cited in comparing and contrasting our method of changing the topological polarization via the uniform soft twisting to their method of altering it via a soliton. [4] is cited as an example of engineering zero modes and self stresses via static structural defects, in contrast to our method of using dynamical nonlinearities. [5] is now cited to compare their method of controlling finite-frequency waves via folding mechanisms to our method of controlling zero-frequency modes via a broader but overlapping family of mechanisms. The discussion of [6] is now expanded in the new section in relation with the creation of domain structures

with a self-stress domain wall.

Referee 1, Point 2:

“I think the authors should set-up a clear distinction between their approach and the approaches in [5], [6], emphasizing not only the different physical nature of the transition mechanisms but also the implications for potential applications (Advantages and disadvantages) of their approach.”

In the third paragraph (starting with “In this Article, we show that, ...”), we compare our design of TTMMs with previously reported tunable mechanical metamaterials, and we summarize the difference into two points. First, the properties in different phases in our design are topologically protected (this differs from [5]). Second, the tuning mechanism we employ is soft and involved no significant stress (as illustrated in our video, which differs from [6]). Rigorous realization of the second point requires “free-hinges” with to resistance to relative rotations between building blocks. Challenges and possible solutions regarding this have been discussed in the second paragraph in the Discussion section.

Referee 2, Point 1:

“As far as I can see, all concepts involved in the work (deformation induced transitions, shifting floppy modes around by deformation, cute videos with lego-like toys etc), as the authors readily acknowledge... The central claim the authors write in italic has certainly been seen in such systems, as well as other floppy mode systems, so perhaps should be de-emphasized.”

We understand this to be a reference to the statement in the abstract that “Our design is based on *transitions between states with distinct topological structures of their phonon bands*, and thus the properties in each state are topologically protected and highly robust against disorder and noise.” We have rewritten that statement to now read

These properties depend on the global structure and, for critically-coordinated Maxwell lattices, the topological structure of the phonon bands, and are hence highly robust against disorder and noise.

We do this to avoid giving the impression that the topologically nontrivial states are a novel result. These topological states have been extensively discussed. However, a practical method to induce transitions between distinct topological states have not been characterized. In the original paper by Kane & Lubensky (Ref. 11) it was shown that different lattice geometries can have distinct and nontrivial topological polarizations. These changes in geometry corresponded to differing material geometry (bond lengths) rather than elastic response of the

lattices to simple external stresses or strains. The transitions discussed in Ref. 11 are purely theoretical and have not been demonstrated in real systems [with the only exception of changing polarization through a nonlinear soliton in a one-dimensional chain (Ref. 14) as correctly pointed out by both referees]. Instead, our approach links changing topological polarization to a class of soft strain that are guaranteed to exist in the lattices, and thus provide a very practical way for switching topological modes in real systems. We have also shown that these changes alter material properties and, in the case of edge stiffness, demonstrated this using the “K’nex” prototype.

Referee 2, Point 2:

“I would be very interested to hear from the authors more about why this is interesting or useful... On the ‘useful’ front, the authors make some somewhat vague references to the wonders of modern material science. In my opinion the sub-field is past this being a suitable justification and more precise identification of applications is necessary.”

We acknowledge that useful applications are desirable and that we did not demonstrate clear, precise applications or practical means of manufacturing bulk materials. We regard our focus on a general phenomenon rather than resting on specific application. Nevertheless, we have endeavored to present more specific applications of our design where it could potentially transform an important area of research, with the addition of the following paragraph:

“Moreover, the ability of a material to both alter its mechanical state in response to its environment and to robustly maintain such a state is characteristic of living tissue at all scales. The TTMMs exhibit the similar property of reversibly and immediately changing surface stiffness, and this opens the door to interesting application in biomedical research. Studies using substrates of various but *fixed* stiffnesses have shown that this affects cell locomotion, inter-cell communication and even, famously, development and differentiation. A substrate composed of a pure, microstructured TTMM could produce surface stiffness *controllably changing* in real time across orders of magnitude, controllably mimicking *in vivo* conditions responding to disease, damage and environmental changes.” (References omitted here.)

Referee 2, Point 3:

“On the interesting front, I see potential in the work, perhaps the authors could make use of the nominal softness of the deformation modes to go to entirely new regimes? New nonlinear excitations? Thermal effects enabled by softness? Any one of these (or other)

concepts that could be enabled by the identification of this region of parameter space would be interesting and give the work some conceptual impact.”

Inspired by this comment, we have indeed incorporated new nonlinear excitations into the manuscript which, as identified above, permit the realization of new regimes (see the new section “creation of domain structure”). We observe that in fact these are generic to the class of materials already identified and as such do not require a new or restricted region of parameter space. We feel that these additions have significantly expanded the conceptual impact of the work.

We agree that thermal fluctuations in such lattices are of great interest. However, this is a rich enough topic to warrant its own paper, and some of the authors are currently preparing just such a manuscript.

Referee 2, Point 4:

“I would also recommend the authors expand their discussion of topological mechanics in the early paragraphs. The field now includes other systems, such as coupled pendula (ETH), spinning top mechanics (Harvard, Chicago), among others.”

We have included a number of new references in our introduction to expand our survey of current research on topological mechanics, including the ones suggested by the referee, towards the end of the paragraph starting “What is central to the understanding ...”

REVIEWERS' COMMENTS:

Reviewer #1 (Remarks to the Author):

Dear Editor,

I consider that the authors have provided a satisfactory answer to my queries. The manuscript has improved its readability, acknowledges previous work in the field properly and includes a clear comparison between their approach and past approaches.

I am hence recommending the manuscript for publication in Nature Communications.

Reviewer #2 (Remarks to the Author):

The manuscript has improved considerably since the first version. The authors now cite many more relevant papers and have made more modest and accurate claims. Regarding claims of applications - It was not my intention to encourage far fetched applications, rather to encourage the authors to either propose real applications or stay away from claiming their work has applications. The analogies to cell compliance they added seems very far fetched and I encourage removing it.

In conclusion, the manuscript has much improved, though, as now more clearly stated, presents modest progress with respect to the published literature. It is in my opinion acceptable for Nature Communications, if marginally so.

We thank the editor and referees for their careful consideration of our manuscript, “Transformable topological mechanical metamaterials”. We are very pleased that a suitably revised version is suitable for publication.

In response to the comment of Reviewer 2, we have removed the paragraph discussing biological cells.